# Rehabilitation and violence-related traumatic brain injury: A scoping review

**Samira Omar**[1☉], **Maria Jennifer Estrella**[2☉], **Muzna Ahmad**[2], **Angela Colantonio**[1,2,3,4], **Jessica Babineau**[5,6], **Vincy Chan**[1,3,4]*

1 Rehabilitation Sciences Institute, University of Toronto, Toronto, ON, Canada, 2 Department of Occupational Science & Occupational Therapy, University of Toronto, Toronto, ON, Canada, 3 KITE Research Institute, Toronto Rehabilitation Institute-University Health Network, Toronto, ON, Canada, 4 Institute of Health Policy, Management and Evaluation, University of Toronto, Toronto, ON, Canada, 5 Library and Information Services, University Health Network, Toronto, ON, Canada, 6 The Institute for Education Research, University Health Network, Toronto, ON, Canada

☉ These authors contributed equally to this work.
* vincy.chan@uhn.ca

**Data Availability Statement:** All relevant data are within the manuscript and its Supporting Information files.

## Abstract

### Objectives

There is a dearth of reviews exploring rehabilitation for violence-related traumatic brain injury (TBI) despite its increasing rates and distinct risk factors and outcomes. The aim of this scoping review is to determine the extent to which rehabilitation is available to and accessed by people who sustain TBI from interpersonal violence.

### Method

Electronic databases (i.e., MEDLINE, Cochrane CENTRAL Register of Clinical Trials, CINAHL, APA PsycINFO, Criminal Justice Abstracts, Applied Social Sciences Index and Abstracts, and Proquest Nursing and Allied Health) and grey literature (i.e., relevant organizations' websites) were systematically searched for peer-reviewed articles and reports that met eligibility criteria. To be included, articles had to report primary findings on rehabilitation interventions that included individuals who sustained a TBI through interpersonal violence. Articles based on the military population were excluded. Data were synthesized through a descriptive numerical summary and qualitative content analysis.

### Results

Fifty-two primary research articles and five grey literature reports were included. Most (77.2%) of these articles described rehabilitation interventions that addressed TBI sequalae without consideration for violence as a mechanism of injury, beyond reporting sample characteristics. Only thirteen articles (22.4%) considered violence-related TBI by investigating the rehabilitation profile (13.8%) or designing programs specifically for survivors of violence-related TBI (8.6%). There was limited attention to social determinants of health across all studies.

**Funding:** This study was supported by the Canada Research Chairs Program (Grant 2019-00019 to AC) and the Ontario Ministry of Health (Grant #725A to AC & VC). The reviews expressed in this publication are those of the authors and do not necessarily reflect those of the Ministry of Health. This study was supported by an inaugural Black Scholars Personnel Award (to SO) funded by Heart & Stroke, Brain Canada, and Canadian Institutes of Health Research Institute of Circulatory and Respiratory Health (CIHR-ICRH). The views expressed herein do not necessarily represent the views of the Minister of Health or the Government of Canada.

**Competing interests:** The authors have declared that no competing interests exist.

## Conclusion

This is the first scoping review to our knowledge that explored the extent and nature of rehabilitation among persons who experience TBI through violence in civilian populations. Priorities for education, research, and practice are identified.

## Introduction

Traumatic brain injury (TBI), defined as "an alteration in brain function or other evidence of brain pathology, caused by an external force" [1], is a significant public health concern and a leading cause of death and disability [2]. Individuals living with TBI experience long-term disability, including persistent limitations with physical and cognitive functioning and mental and emotional well-being which can last for several years and impact their quality of life [3]. The most common causes of TBI are falls, blunt trauma accidents, vehicle-related injuries, and assault or violence [4].

Recently, attention has been directed towards violence as a mechanism of brain injury, which draws on an important intersection affecting largely women survivors of intimate partner violence (IPV) [5]. This classification of violence is however a subset of the substantial portion of the population of persons impacted by violence, namely interpersonal violence, which represent an underserved group with unmet needs, unique injury characteristics, and functional outcomes. The World Health Organization defines interpersonal violence as "the intentional use of physical force or power, threatened or actual, against. . .another person, or against a group or community that either result in or has a high likelihood of resulting in injury, death, psychological harm, maldevelopment or deprivation" [6]. This definition captures a broad range of persons impacted by violent actions and intent resulting in TBI, including those perpetuated by family or partner violence and community violence between friends, acquaintances, or strangers. From here on, interpersonal violence is used to capture the broader range of violence as a mechanism of TBI.

Interpersonal violence is a significant public health concern [6] and survivors of violence-related TBI represent a distinct clinical group requiring greater attention. Survivors of violence-related TBI tend to have poorer functional outcomes after being discharged from inpatient rehabilitation compared to individuals who sustain TBI from other causes [7]. Interpersonal violence is also one of the leading causes of TBI amongst racial and ethnic minorities, particularly Black and Indigenous populations. These populations are disproportionately impacted, endure more severe injuries [8, 9], have limited access to rehabilitation [10, 11], and experience poor life satisfaction and quality of life [12] as a result of their TBI. These health disparities as well as their access to rehabilitation are shaped by a multitude of social determinants of health (SDoH), including racism, education, housing, income, living environment, and life opportunities.

Rehabilitation is broadly defined as "a set of interventions designed to optimize functioning and reduce disability in individuals with health conditions in interaction with their environment" [13]. Despite having positive effects on functional recovery [14], symptoms, participation, and quality of life [15], TBI rehabilitation has been criticized for utilizing a one-size-fits-all approach that lacks consideration for nuances in the demographic profile of injury based on gender, race, and mechanism of injury [2]. This conflicts with the need for unique rehabilitation programs and interventions that address the injury-related nuances of violence-related TBI [7, 16]. However, it is not yet known to what extent rehabilitation, including the types of

rehabilitation interventions, account for the unique needs of survivors of violence-related TBI. To date, reviews of the literature have focused on comparing symptoms associated with violence-related TBI with other mechanisms of injury [17], reporting on specific forms of interpersonal violence (e.g., intimate partner violence) [18], mapping the scope of the literature [19], and providing a comprehensive overview of the risk factors, outcomes, and service implications of violence-related TBI [9].

The differing functional outcomes and demographic profile of survivors of violence related TBI [9, 16] are pressing issues that need to be addressed to achieve equitable rehabilitation care. Additionally, rehabilitation for persons experiencing chronic health conditions such as TBI is a priority for the 21st century and one of the reasons the WHO launched the 2030 Rehabilitation Goals—to optimize functioning and participation through relevant programs and interventions to address unmet needs [20]. This scoping review aims to address these knowledge gaps and determine to what extent rehabilitation is available to and accessed by people who experience TBI from interpersonal violence, with the goal of identifying priorities for rehabilitation practice and research. Given what is known regarding the demographic profile of survivors of violence-related TBI, special attention was drawn to intersecting SDoH, including sex, gender, race, ethnicity, age, and other sociodemographic characteristics in addressing the objectives of the review.

## Methodology

This scoping review closely follows the methodology of a series of scoping reviews on TBI in underserved populations [21, 22] and is developed under the methodological guidance of Arksey and O'Malley [23] and Levac et al., [24] which followed six steps described below. As part of the reporting of this scoping review, the Preferred Reporting Items for Systematic reviews and Meta-Analyses Extension for Scoping Reviews (PRISMA- ScR) checklist were used [25].

### Identifying the research question

This scoping review aimed to answer the question, "what is the extent and nature to which rehabilitation, including the types of rehabilitation interventions, is available to or accessed by people who are experiencing TBI through mechanisms of violence?" This question informed the development of the search strategy, selection of relevant literature, charting of the data, analysis of the findings, and reporting of the results. Table 1 operationally defines key terms such as TBI, rehabilitation, and violence, which have been used to guide all steps of the review.

### Identifying the relevant studies

The search strategy was informed by previous scoping and systematic reviews [21, 22] and was developed through an iterative process in collaboration with an information specialist and members of the research team who hold content and research expertise in the areas of TBI, violence, and rehabilitation. Three concepts were used to build and finalize the search strategy (violence, TBI or cognitive impairment, and rehabilitation). The final search strategy consisted of combining these concepts in the following way: [(violence) AND (TBI or cognitive impairment) AND (rehabilitation)] OR [(rehabilitation) AND (violence-related TBI)]. To identify relevant peer-reviewed literature, the search strategy for the MEDLINE® ALL (in Ovid, including Epub Ahead of Print, In-Process & Other Non-Indexed Citations, Ovid MEDLINE(R) Daily) database was first developed and then translated to the following seven databases: Embase and Embase Classic (Ovid), Cochrane CENTRAL Register of Clinical Trials (Ovid), CINAHL (EBSCO), APA PsycINFO (Ovid), Applied Social Sciences Index and Abstracts (Proquest), Criminal Justice Abstracts (EBSCO), and Nursing and Allied Health

Table 1. Definitions for traumatic brain injury (TBI), rehabilitation, and violence.

| Concept | Definition |
|---|---|
| TBI | "An alteration in brain function, or other evidence of brain pathology, caused by an external force" [1] |
| Rehabilitation | "A set of interventions designed to optimize functioning and reduce disability in individuals with health conditions in interaction with their environment" [13]<br>Healthcare providers/professional disciplines identified in clinical practice guidelines for rehabilitation for TBI [26, 27]:<br>• Neuropsychologist and psychometrist<br>• Nurse<br>• Nutritionist<br>• Occupational therapist<br>• Physician and/or physiatrist<br>• Physiotherapist<br>• Psychologist with expertise in behavioural therapy<br>• Rehabilitation support personnel<br>• Social worker<br>• Speech-language pathologists<br>• Therapeutic recreationist |
| Violence | "The intentional use of physical force or power, threatened or actual, against oneself, another person, or against a group or community, that either results in or has a high likelihood of resulting in injury, death, psychological harm, maldevelopment or deprivation." |
| Interpersonal violence | Interpersonal violence is divided into two subcategories [28]:<br>• Family and intimate partner violence–that is, violence largely between family members and intimate partners, usually, though not exclusively, taking place in the home.<br>• Community violence–violence between individuals who are unrelated, and who may or may not know each other, generally taking place outside the home. The former group includes forms of violence such as child abuse, intimate partner violence and abuse of the elderly. The latter includes youth violence, random acts of violence, rape or sexual assault by strangers, and violence in institutional settings such as schools, workplaces, prisons and nursing homes. |

(Proquest). There were no language or date restrictions placed on the search strategy. However, a search filter was applied to exclude animal studies. Searches were conducted across the different databases in January 2023.

This scoping review also included grey literature described as reports from brain injury, violence, and rehabilitation organizations. These reports were obtained from organizations' websites by manually searching through the websites and consulting with stakeholders. End-Note X8.2 was used for deduplication and reference management, and Covidence was used for deduplication and screening. For more details about the search strategy for each of the seven databases and the list of organizations searched to identify grey literature reports, please see S1 File.

## Selecting the studies

All peer reviewed articles as well as grey literature reports identified from the search were subject to the following inclusion criteria:

1. Describe and/or document rehabilitation interventions and/or services provided by healthcare providers or professional disciplines as defined in Table 1

2. Include individuals who sustained a TBI through interpersonal violence as defined in Table 1

3. Report primary research findings

The following articles were excluded:

1. Dissertations, conference proceedings, and articles that are narrative, commentaries, or describe a theory/framework without presenting primary research findings

2. Articles looking at the broader brained injury/cognitive impairment population without specific mention of TBI

Using Covidence and the criteria outlined above, two reviewers screened all English and non-English language titles and abstracts independently. Articles were considered for full-text review at this stage if they included persons with TBI or acquired brain injury or cognitive impairment and rehabilitation. Scoping and systemic reviews were also considered at this stage if they met these inclusion criteria. Eligibility for all non-English language articles was assessed using the published English abstract. A pilot screen of 20 titles and abstracts was conducted between the two reviewers until a minimum 80% agreement was reached. The resulting average agreement was 95.9% (kappa 0.880) at the title and abstract screen for English and 96.3% (kappa 0.867) for non-English language articles.

At the level of full-text screening, each article was independently screened by two of three reviewers. Any primary research articles identified from the reference lists of scoping and systematic literature reviews included at the level of title and abstract screening were extracted and independently screened based on the above criteria. Similar steps were followed to extract grey literature reports. All non-English language articles were translated into English using Google or DeepL Translate. A pilot screen of 10% of eligible full-text articles was conducted until a minimum 80% agreement was reached between the two reviewers. The resulting average agreement on the full-text screen was 95.1% (kappa 0.690) for English and 96.4% (kappa 0.650) for non-English language articles.

## Charting the data

A data charting form was created to extract relevant information to answer the research question. Data were charted independently by two reviewers and peer-reviewed by two reviewers. Any discrepancies in the charting of the data were discussed and resolved by consensus with a third reviewer. The charting table is presented in S2 File.

## Collating, summarizing, and reporting the results

Findings from this scoping review are presented using a descriptive numerical summary of the included studies, which are reflected in the data charting table, as well as a qualitative content analysis to describe the extent and nature to which rehabilitation is available to and accessed by people who experience TBI through interpersonal violence. The findings from the quantitative and qualitative content analysis were used to answer the research question and identify research and practice priorities for survivors of violence-related TBI.

## Consultation

Preliminary findings were shared with members of a Program Advisory Committee (PAC), who bring forward several forms of expertise. The PAC includes front-line staff and service providers in the violence and brain injury sectors, healthcare professionals who provide care or support for persons with TBI and survivors of violence, as well as researchers and trainees who conduct research on the intersections of violence, brain injury, and rehabilitation. The PAC provided feedback regarding the preliminary findings of the review, which was then integrated in the discussion of the findings.

PRISMA 2020 flow diagram for new systematic reviews which included searches of databases, registers and other sources

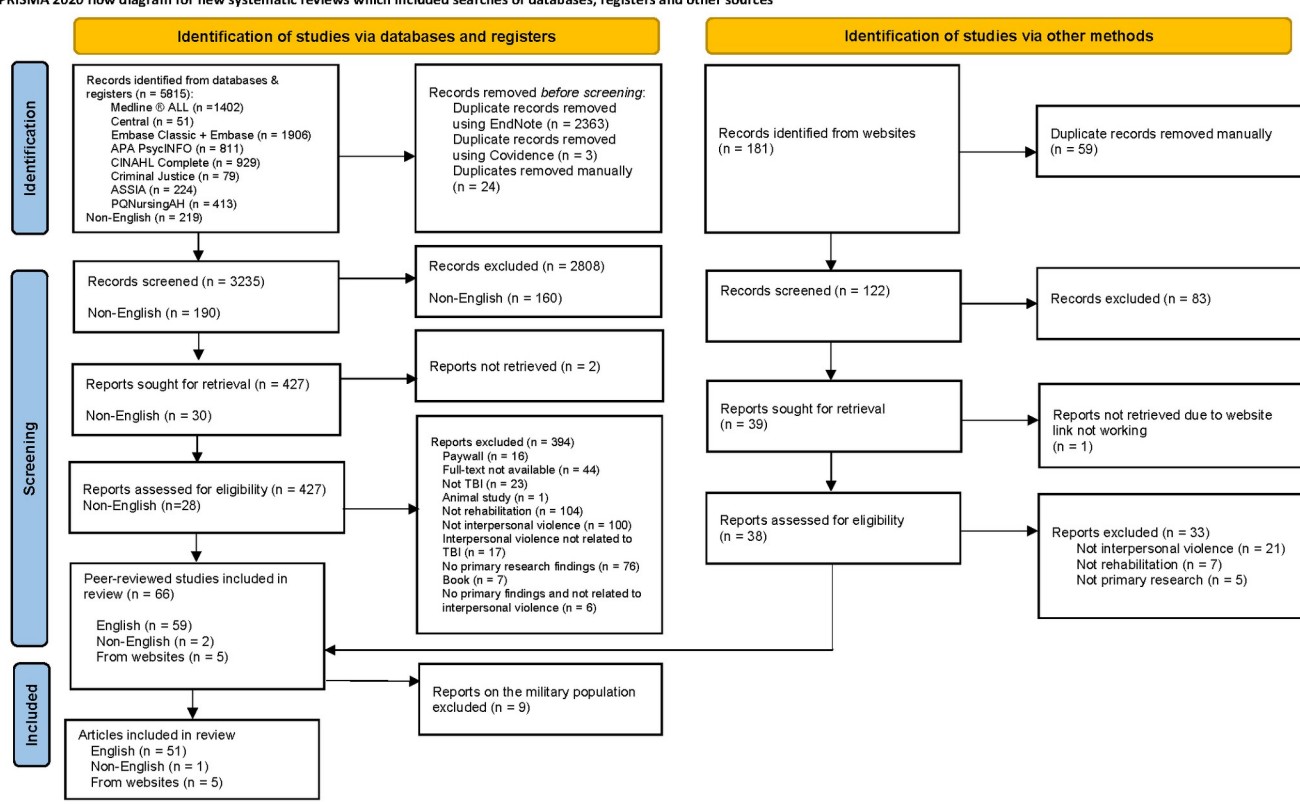

*From:* Page MJ, McKenzie JE, Bossuyt PM, Boutron I, Hoffmann TC, Mulrow CD, et al. The PRISMA 2020 statement: an updated guideline for reporting systematic reviews. BMJ 2021;372:n71. doi: 10.1136/bmj.n71. For more information, visit: http://www.prisma-statement.org/

**Fig 1. PRISMA flow diagram.**

## Results

Fifty-two primary research articles (51 English, 1 non-English language) and 5 grey literature reports were included from a total of 3425 citations (190 in non-English language) after removing duplicates. We also identified and further excluded nine articles that were focused on the military population. These articles were excluded, as evidence suggests that military veterans are a distinct group that may have very different experiences of violence and rehabilitation compared to individuals who sustain a violence-related TBI through other contexts [29]. The PRISMA Flow Chart (Fig 1) shows the study selection process, including reasons for exclusion. Table 2 presents the characteristics of the studies included in this review and S2 File presents the charting table.

The majority of the studies included in this review were conducted in the United States (43.9%) and published between 2016 and 2022 (45.6%). Five (8.8%) of these studies were RCT's and three were specifically qualitative (5.3%) by design. Rehabilitation interventions were primarily delivered in inpatient (40.4%) or outpatient settings (21.1%). The most common type of violence, as described by the authors of the included studies, were assault (52.6%), gun violence (22.8%), and IPV (10.5%). Only four studies described how the assault occurred (i.e., through a fist fight, with an object such as a bat or low caliber projectile). Five studies (8.8%) focused on child abuse and five (8.8%) reported violence as a mechanism of injury but did not specify the type of violence.

**Table 2. Study characteristics (N = 57).**

| Characteristics | N (%) |
|---|---|
| **Year of publication** | |
| 1975–1985 | 2 (3.5) |
| 1986–1995 | 1 (1.8) |
| 1996–2005 | 12 (21.1) |
| 2006–2015 | 16 (28.1) |
| 2016–2022 | 26 (45.6) |
| **Country of study** | |
| United States | 25 (43.9) |
| Canada | 5 (8.8) |
| United Kingdom | 6 (10.5) |
| Australia | 5 (8.8) |
| **European Countries** | 8 (14.0) |
| Ireland | 1 (1.8) |
| France | 1 (1.8) |
| Norway | 1 (1.8) |
| Germany | 1 (1.8) |
| Poland | 1 (1.8) |
| Romania | 1 (1.8) |
| Russia | 1 (1.8) |
| Denmark | 1 (1.8) |
| **Middle Eastern Countries** | 5 (8.8) |
| Saudi Arabia | 1 (1.8) |
| Israel | 2 (3.5) |
| Turkey | 2 (3.5) |
| South Africa | 2 (3.5) |
| Taiwan | 1 (1.8) |
| **Study design** | |
| Before-after, no control | 6 (10.5) |
| Case report/case series | 22 (38.6) |
| Cohort | 19 (33.3) |
| Cohort, case study | 1 (1.8) |
| Cohort, case study, qualitative | 1 (1.8) |
| Qualitative | 3 (5.3) |
| Randomized Controlled Trial | 5 (8.8) |
| **Age[a]** | |
| Infant (<1) | 4 (7.0) |
| Toddler (1–3 years) | 2 (3.5) |
| Adolescent (13–17 years) | 2 (3.5) |
| Adult (18–64 years) | 50 (87.7) |
| Older Adult ($\geq$ 65) | 7 (12.3) |
| "Adult, parent, child" (age not specified) | 1 (1.8) |
| **Sex and gender** | |
| Males/men > Females/women[b] | 24 (42.1) |
| Females/women > Males/men[c] | 7 (12.3) |
| Females or Women only | 6 (10.5) |
| Males or Men only | 16 (28.1) |
| Not reported | 4 (7.0) |

(*Continued*)

**Table 2.** (Continued)

| Characteristics | N (%) |
|---|---|
| **Race and ethnicity** | |
| Not reported | 35 (61) |
| White participants only | 1 (1.8) |
| Included racial and ethnic minority groups | 21 (36.8) |
| Black/African | 11 (19.2) |
| Asian | 4 (7.0) |
| Middle Eastern | 4 (7.0) |
| Indigenous populations | 3 (5.3) |
| Hispanic/Latino | 9 (15.8) |
| Not specified | 8 (14.0) |
| **Location of Rehabilitation** | |
| Inpatient | 23 (40.4) |
| Outpatient | 12 (21.1) |
| Hospital (emergency department) | 1 (1.8) |
| Hospital (not specified) | 6 (10.5) |
| Inpatient and outpatient | 2 (3.5) |
| Shelter or residential | 2 (3.5) |
| Community | 2 (3.5) |
| Not specified | 9 (15.8) |
| **Type of violence**[b] | |
| Assault | 30 (52.6) |
| Intimate Partner Violence | 6 (10.5) |
| Gun violence | 13 (22.8) |
| Bomb violence | 1 (1.8) |
| Child abuse | 5 (8.8) |
| Blunt force | 1 (1.8) |
| Violence (not specified) | 5 (8.8) |

[a]Each study may focus on more than one age group; thus, the total N and % will not equal to 100%.

[b]More males/men than females/women participants

[c]More females/women than males/men

[d]Study documented more than one type of violence sustained by participants; thus, the total N and % will not equal to 100%.

There was limited reporting and description of intersecting SDoH in the 57 studies included in the review. Study samples in 40 studies included mostly males or men (42.1%) or only males or men (28.1%); only six articles included only females or women (10.5%). Four (7.0%) did not report any information about sex and/or gender. Similarly, only 22 (38.6%) articles reported information regarding race and ethnicity, with 21 studies (36.8%) including participants who belonged to racial and ethnic minority groups, 35 studies (61.4%) did not report on race and ethnicity, and 8 (14.0) categorized racial and ethnic minority groups as "other" or "minority" without specifying participants' race or ethnicity. Only four studies provided an intersectional analysis that was stratified by race and gender and accounted for racial disparities in rehabilitation outcomes for persons with violence-related TBI [39–41, 44]. Notably, 43 studies (75.4%) reported information about the participants' sociodemographic characteristics, including employment, level of education, marital status and socio-economic status. However,

only 5 (8.8%) studies reported intersections with homelessness and the criminal justice system (CJS) [30–34].

Two categories of articles were identified following content analysis: (1) articles that described and/or documented rehabilitation interventions that addressed TBI sequalae without consideration for violence as a mechanism of injury (n = 44), and (2) articles that considered violence-related TBI (n = 13) by investigating the rehabilitation profile and outcomes of violence-related TBI (n = 8) and designing rehabilitation interventions for survivors of violence-related TBI (n = 5). Detailed findings for each category are presented below.

## Rehabilitation that addresses TBI sequalae without consideration for violence as a mechanism of injury (N = 44)

Forty-four of the 57 articles (77.2%) documented and/or described rehabilitation interventions that addressed TBI sequalae without consideration for violence as a mechanism of injury, beyond reporting sample characteristics (i.e., the number of participants who sustained their TBI through violence). Rehabilitation interventions were delivered primarily in hospital settings, including inpatient and outpatient units, and a few in the community [34, 35] and in shelter or residential settings [36]. Healthcare professionals involved in delivering the rehabilitation included physical therapists or physiotherapists, occupational therapists, speech therapists, art therapists, blind rehabilitation specialists, nurses, rehabilitation assistants, optometrists, and psychologists.

Twenty-one of the 45 articles (46.7%) documented but did not further describe the rehabilitation intervention, setting, or rehabilitation professionals seen by survivors of violence-related TBI. Examples of rehabilitation in these articles included inpatient or acute and outpatient rehabilitation, neurorehabilitation, and various therapies, such as intensive physical therapy and psychiatric treatment. In contrast, 23 articles documented and/or described rehabilitation that targeted specific outcomes, such as sensory input and sensory processing, behavioural, systemic or biological, cognitive, psychological, and motor functioning as well as functional or related to day-to-day support or support for life activities (e.g., activities of daily living, work, community living, and social participation). Specific examples of rehabilitation that addressed TBI sequalae related to sensory input and sensory processing included visual restitution training, scanning therapy, and auditory processing program, while rehabilitation interventions that addressed behavioural challenges associated with TBI included anger self-management training and neurorehabilitation program for aggression. Rehabilitation in all 45 articles focused on individuals with TBI in general, and no considerations for violence as a mechanism of injury were reported.

## Rehabilitation that considered violence-related TBI (N = 13)

A total of 13 articles (22.4%) documented and/or described rehabilitation that considered violence-related TBI by (1) investigating the rehabilitation profile (i.e., sociodemographic profile or outcomes) of survivors of violence-related TBI (N = 8) and (2) designing rehabilitation programs for survivors of violence-related TBI (N = 5).

Eight articles investigated the rehabilitation profile of survivors of violence-related TBI by comparing the sample of participants who experienced violence-related TBI to those who sustained TBI through other mechanisms of injury. In these articles, rehabilitation was delivered in inpatient rehabilitation settings [16, 37, 38], a specialized TBI acute rehabilitation unit [39], a level 1 trauma center [40], and a pediatric hospital and outpatient clinic [41]. Only one study specified what type of rehabilitation program or services patients received in a rehabilitation unit such as physical therapy and occupational therapy [42]. It is worth emphasizing that

consideration of violence-related TBI in these articles is limited to a description of survivors' sociodemographic profile and outcomes. It was not reported if or how violence as a mechanism of injury was considered in the rehabilitation process.

Of these eight studies, three described the sociodemographic profile of survivors of violence-related TBI. Often, survivors of violence-related TBI were unmarried [38, 39], young males or men from a racial background other than White [38–40], and with a history of substance use [39, 40]. These individuals also often had a shorter length of stay in acute care [39] and accessed services through government (e.g., Medicaid) funding [41].

All eight studies investigated rehabilitation outcomes for the population. Four of the eight studies compared rehabilitation outcomes between violent and non-violent TBI groups, such as TBI due to motor vehicle collisions and falls [16, 37, 38, 40]. Differences for survivors of violence-related TBI included significantly lower scores on reintegration to normal living which continued over 3 to 6 months post-discharge [16]; small significant differences in functional improvements over time [38], and greater likelihood to be discharged home than to inpatient rehabilitation [40]. However, the latter was reported to possibly be more related to the type of injury deficit and services needed rather than the mechanism of injury. One study investigated the rehabilitation outcomes of bombing survivors specifically and discussed the challenge of short-term rehabilitation stays, long-term disability, and limited funding for ongoing rehabilitation [43]. Three studies focused on pediatric survivors of abusive head trauma. While one article noted no significant differences in functional impairments post-discharge [37], two reported significant neurological impairments (e.g., epilepsy, motor, attention and visual deficits, sleep disorders, language abnormalities, and behavioural problems) [42] and regressions in learning, fine motor, and expressive language [41] after several years of abusive head trauma.

Five studies described rehabilitation specifically designed for survivors of violence-related TBI. Four of the five studies focused on IPV resulting in TBI [31, 35, 44, 45] and one on child abuse [46]. These rehabilitation interventions were delivered in an inpatient setting [46], a non-profit facility in a metropolitan area [31] and non-profit community based agencies supporting survivors of domestic violence [45] and persons with brain injuries [35]. A multidisciplinary team was often involved; specific providers included physicians [45, 46], neurologists [44], physiatrist [44], occupational therapists, [31, 44, 46], physical therapists, speech language pathologists [44, 46], social worker [31], and violence impact coordinator [35]. These rehabilitation interventions focused on ensuring safety and preventing violence [31, 44, 46]; regaining or improving self-regulation and control [31, 44, 45]; and addressing violence- and/or TBI-related challenges [31, 35, 44, 46].

## Ensuring safety and preventing violence

Three studies reported on safety planning and violence prevention, with two in the context of IPV. Safety planning and violence prevention were particularly important to individuals who were still connected to or residing with the perpetrator of the abuse. For example, critical questions in an article on feminist psychotherapy were centred on helping the individual avoid physical violence and leave the relationship [44]. Individuals who received feminist psychotherapy benefitted from therapy by implementing suggestions for safety. Examples of these suggestions included moving into a relative's home during therapy and not revealing the location of a new workplace to the perpetrator.

Safety was also an identified need in an initial needs assessment that was part of an occupational therapy (OT) intervention for survivors of IPV who have sustained a TBI [31]. Safety in this intervention encompassed not only safety planning but also community safety,

particularly for survivors who were also experiencing homelessness. Women were taught skills related to these two domains. Examples included identifying safe locations and important documents and items to take, recognizing the signs of a worsening violent incident, and creating a plan where neighbours could call the police if they hear sounds of violence. Community safety skills included identifying signs of danger on the street or in shelters, avoiding revictimization and understanding the link between substance use and assault.

Finally, one article reported on violence prevention in the context of child abuse and the role of OTs in providing education to the family and the community. Although safety planning was not explicitly mentioned in the study, mandatory strategies to ensure the child's safety were reported and included determining a legal guardian, monitoring all visits by family, and providing family education and training as rehabilitation is ongoing, to all caregivers, including suspected abusers [46].

**Regaining or improving self-regulation and control.** Three rehabilitation interventions emphasized self-regulation and control among survivors of violence-related TBI [31, 44, 45]. These interventions included biofeedback, neurofeedback, and an OT intervention that incorporated a stress and anger management component. Biofeedback involves the use of an equipment that provides immediate feedback to help individuals gain control of bodily functions, such as muscle relaxation, heart rate, and temperature [44], while neurofeedback uses qEEG to regulate electrical activity in different areas of the brain [45]. The latter trains the brain in working effectively and ensuring healthy patterns in brainwaves, which can then lead to improved cognitive performance and emotional self-regulation. The goal of these interventions is to enable individuals to make intentional changes on bodily functions and ultimately empower them to gain greater control of their lives. Clients who received neurofeedback and biofeedback showed improvements in mental health symptoms (e.g., depression, anxiety, and post-traumatic stress disorder) and neuropsychological function in part due to increased control over their body [44]. Finally, the OT intervention involved skills training related to anger management and stress reduction. While there was no explicit mention of self-regulation or control as a goal or outcome, survivors of IPV reported valuing these two intervention areas the most; they reported that the techniques helped them feel good about themselves, as they were able control their anger and stress levels and respond to others better [31].

**Addressing violence- and/or TBI-related challenges.** Four of the five rehabilitation interventions focused on addressing violence-related and/or TBI-related challenges. The goals of each program varied and included leaving abusive situations or working on unresolved issues that stemmed from a history of abuse [44]; enabling independence and participation in areas of life through skills training [31] or immediate access to supports [35]; and remediating functional impairments and facilitating normal development in children who experienced abuse [46]. Notable aspects of the rehabilitation that were reported to be beneficial for violence-related TBI included breaking down goals while matching an individual's functional level [31], navigation support [35], and longer duration of the rehabilitation [31, 35]. Aspects of the rehabilitation interventions that were beneficial for TBI but were not directly related to violence as a mechanism of injury included adaptations and compensatory strategies to accommodate TBI impairments [35], and individualized treatment plans [31, 35, 44].

Breaking down goals while matching an individual's functional level and navigation support were critical in helping survivors feel less overwhelmed with working on their goals and accessing supports. Breaking down goals or steps in learning new skills while matching an individual's functional level was cited as one of the most important aspect of the OT intervention that involved skills training in various areas of life for survivors of IPV [31]. This aspect of the OT program was based on the premise that survivors of abuse experience challenges with cognition (e.g., problem solving, decision-making, and planning) that hinder them from

leaving their abusive situation. Participants of the OT intervention reported that the intervention aided them in taking "baby steps" to learn new skills that were necessary to achieve their desired goals. Breaking down goals into smaller tasks and making sure they were tailored to the individual's functional level made the goals feel less overwhelming, leading to experiences of success and feelings of independence and accomplishment. Similarly, navigation support or assistance in accessing supports through a violence impact program was necessary, even when there is an option to self-refer [35]; For individuals dealing with brain injury due to trauma, accessing services can be a complicated and challenging process as they experience cognitive impairments alongside other brain injury challenges (e.g., depression and lack of initiation).

The length of time for rehabilitation was also an important consideration in two studies that focused on supporting survivors of IPV and TBI [31, 35]. It was noted in these studies that survivors of brain injury due to abuse may require longer periods of time before they can make small changes, and therapy will involve a series of gains and relapses rather than a linear recovery [31]. Symptoms of TBI such as emotional exhaustion, memory impairments and lack of follow through can also hinder survivors from effectively receiving support. A longer duration of rehabilitation or long-term supports are thus important to ensure that there is adequate time for survivors to achieve their goals [35].

Three studies highlighted the importance of adaptations or compensatory strategies for TBI [31, 35, 44] in the delivery of the intervention for survivors of violence-related TBI. The delivery of feminist psychotherapy involved repetition (i.e., asking the individual to repeat the therapist's explanations) to ensure comprehension, written materials to supplement verbal information, and review of written materials to ensure understanding given challenges with reading [44]. Written materials also included schedules to help individuals manage their appointments. Verbal positive reinforcement was used as well as negative reinforcement to reduce tangential comments. Compensatory strategies in the OT intervention included repeated practice and adaptive devices [31], and the violence impact program involved the violence impact coordinator working with survivors to develop a support plan that included compensatory strategies to assist them in managing cognitive, emotional, and behavioural challenges [35].

Finally, three studies acknowledged the importance of individualized treatment plans in supporting survivors of violence-related TBI. Successful neuropsychologically-informed feminist psychotherapy required culturally relevant assessments and goals that are mutually agreeable to the therapist and the individual receiving services [44]. Survivors who received the OT intervention identified the client-centredness of the intervention (i.e., the intervention is specific to survivors' personal goals) as one of its most important qualities [31], and the violence impact program emphasized self-identified goals and needs in service delivery and the client-centred approach in their service model [35]. It is worth noting that while individualized treatment plans were a common aspect of these studies, the studies did not articulate why they would be important or beneficial to the rehabilitation of survivors of violence-related TBI.

## Discussion

This scoping review is the first, to our knowledge, to explore the extent and nature to which rehabilitation is available to and accessed by people who experience TBI through interpersonal violence as reported in the literature. While 57 studies (52 primary research articles, 5 grey literature reports) were included in the review, only 13 articles explicitly considered violence-related TBI by investigating the rehabilitation profile (n = 8) and designing rehabilitation specifically for survivors of violence-related TBI (n = 5). The following priorities for practice and research were identified: (1) education regarding the rehabilitation profile of violence-related

TBI; (2) research regarding the needs of survivors of violence-related TBI through mechanisms other than abuse; and (2) explicit considerations for intersecting SDoH in the design and delivery of rehabilitation for this population.

## Education regarding the rehabilitation profile of violence-related TBI

Findings regarding the profile of survivors of violence-related TBI are consistent with the findings of an earlier scoping review on brain injury as a result of violence [9]. Survivors of violence-related TBI were often unmarried young males or men from a racial background other than White who have a history of substance use [38–40]. They were more likely to use government-related funding, have a shorter length of stay in acute care and rehabilitation [39, 40], and poorer outcomes with regard to reintegration to normal living [16]. However, this review adds to the evidence base by including studies on the pediatric population. While one study showed no differences in terms of functional impairment between children who experienced abusive head trauma and those who did not [37], studies that examined the long-term rehabilitation and outcomes of this group noted significant neurological impairments and challenges in learning, fine motor and expressive language [41, 42].

These findings substantiate earlier findings regarding the unique needs of this population, and given rehabilitation's current one-size-fits-all approach, advocates for increased education and awareness regarding the distinct profile and outcomes associated with violence-related TBI. Several studies have reported on the potential benefits of education on TBI rehabilitation, including enabling providers in recognizing TBI signs and symptoms [19], addressing negative views regarding TBI [19, 47] and facilitating detection and access to appropriate services [22, 48]. In the context of this work, such education is a step towards rehabilitation providers recognizing and being better equipped to address the challenges experienced by survivors of violence-related TBI and modifying existing rehabilitation interventions to accommodate their unique needs.

## Research regarding the rehabilitation needs of survivors of violence-related TBI through mechanisms other than abuse

This review identified few research articles that described rehabilitation for survivors of violence-related TBI. While several conceptualizations of violence (e.g., assault to the head, fights, and terror attacks) were reported, the limited number of articles that described rehabilitation for violence-related TBI focused on survivors of abuse. This finding suggests that the rehabilitation needs of individuals who sustained their TBI through other circumstances of violence are either unknown, not described in the published literature, or are not being addressed in TBI rehabilitation.

Worth noting are aspects of rehabilitation interventions that were reported to be important and/or beneficial for survivors of violence-related TBI, specifically abuse: safety planning and violence prevention, regaining or improving self-regulation and control, breaking down goals and matching an individual's functional level, navigation support and longer duration of rehabilitation. First, safety and violence prevention was relevant not just in the home for survivors who still reside with the perpetrator of the abuse, but also in the community, especially for survivors experiencing homelessness [31, 44]. However, we cannot confirm whether this is an important consideration in the pediatric population, as there was only one study on child abuse included in the review. Second, interventions focused on improving and regaining self-regulation and control were beneficial for improving IPV survivors' mental health symptoms, reducing stress, managing anger, and attaining stability and well-being [44, 45]. However, the connection between violence as a mechanism of injury and self-regulation and control as a

goal was not clearly reported in these studies. Third, breaking down goals while matching an individual's functional level [31] as well as navigation support [35] were beneficial in helping survivors feel less overwhelmed and in making tasks more manageable. These aspects of the rehabilitation were intended to address cognitive impairments from TBI and/or trauma that make accessing services a complicated and challenging process and that hinder survivors from leaving abusive situations [31, 35]. Finally, a longer duration of rehabilitation should be considered, as often a series of gains and relapses rather than linear recovery is expected for survivors of IPV-TBI. As such, they may require longer periods of time before they can make small changes [31]. These aspects of the rehabilitation interventions, while intended for survivors of abuse, could be integrated and further investigated in existing or future rehabilitation interventions for the broader population of individuals who sustain TBI through violence.

Also worth noting and integrating in rehabilitation interventions are aspects of rehabilitation interventions for violence-related TBI that have already been identified as ways of adapting rehabilitation interventions for other groups experiencing TBI or individuals with TBI more broadly. These include adaptations or compensatory strategies for TBI and individualized treatment plans [21, 22, 49]. These are worth highlighting as these aspects of the rehabilitation interventions were noted to also be beneficial for survivors of violence-related TBI. However, the studies did not elaborate on how such aspects would be beneficial for this group in particular. Further research is needed to explore the rehabilitation needs of individuals with lived experience of TBI that resulted from other mechanisms of violence other than abuse. Such research will aid in the design and development of programs for the broader population of individuals who sustain TBI through violence.

## Explicit considerations for intersecting social determinants of health in rehabilitation for violence-related TBI

Finally, there is a need to explicitly consider intersecting SDoH in the rehabilitation process (i.e., access, service delivery and outcomes) for violence-related TBI. This review found that studies did not always report important demographic characteristics such as sex, gender, race, ethnicity, and socio-economic status, and only three considered intersections with health disparities, such as experiences with homelessness and involvement with the CJS. Even when sociodemographic characteristics were reported, there was no information regarding how or if they were considered in the rehabilitation process. It is crucial that SDoH is considered, given that violence-related TBI has been found to be associated with a particular demographic profile (i.e., unemployed with a history of substance use and CJS involvement and from a racial and ethnic minority group) [9]. Intersecting SDoH could also create more barriers to rehabilitation. For example, illiteracy [34], transportation challenges [50], and lack of resources, including finances and cost of treatment [43, 51] were some of the barriers to rehabilitation identified in included studies. There is also evidence regarding the differential rehabilitation needs and experiences of racial and ethnic minorities, such as Black persons [11] and Indigenous populations [52] and discrepancies in outcomes for individuals from culturally and linguistically diverse backgrounds [53]. It is clear that intersecting SDoH shape rehabilitation access, experiences, and outcomes, and thus require consideration in the design of interventions or adaptations for existing interventions for survivors of violence-related TBI. While none of the included studies provided any information about how to modify rehabilitation for persons with violence-related TBI based on sociodemographic characteristics such as race, ethnicity, or gender, three of the five studies that provided rehabilitation for survivors of violence-related TBI noted the importance of culture in assessment and treatment. These studies noted the importance of setting goals that are congruent with individuals' cultural values [31],

selecting culturally relevant assessments [44], and increasing culturally competent services for survivors. Addressing these priorities is a critical step toward addressing inequities in healthcare for individuals with TBI. Important to note is the lack of consideration for a trauma-informed approach to rehabilitation given intersecting SDoH and the need for trauma-informed rehabilitation care, particularly for this clinical population of individuals who sustained a violence-related TBI.

We acknowledge the following limitations of this scoping review. First, this review may be subject to publication bias. While we aimed to minimize this by searching for non-English language articles, the websites that we searched for grey literature were all in English; as such, we may have missed potentially relevant non-English grey literature reports. Second, because we only included published reports, rehabilitation programs that may be relevant but not formally reported were not captured. Lastly, we are unable to comment on the efficacy of the rehabilitation programs and interventions unless reported in the article, as this is outside the scope of this review. However, we acknowledge that this information is critical and warrants future investigation.

## Conclusion

This scoping review is the first, to our knowledge, to explore the extent and nature to which rehabilitation interventions are available to and accessed by individuals who sustain TBI due to violence in a range of civilian contexts. More than half of the articles did not consider violence as a mechanism of injury in the rehabilitation process, and the few that did considered violence by investigating the profile of survivors of violence-related TBI and developing rehabilitation interventions for survivors of abuse, primarily women and children. Immediate next steps to advance rehabilitation for this population include education regarding the rehabilitation profile of survivors of violence-related TBI, research regarding the needs of individuals who sustain violence-related TBI through mechanisms other than abuse, and explicit considerations for intersecting SDoH in the design and delivery of rehabilitation interventions for this underserved group.

## Supporting information

**S1 Checklist. Preferred Reporting Items for Systematic reviews and Meta-Analyses extension for Scoping Reviews (PRISMA-ScR) checklist.**
(PDF)

**S1 File. Search strategy.**
(PDF)

**S2 File. Charting table.**
(PDF)

## Acknowledgments

We would like to acknowledge Elias Espinosa for assisting with screening titles and abstracts and Mohammad Adam for peer-reviewing the data extracted and assisting with analysis of study characteristics. We would also like to further acknowledge the Program Advisory Committee of the Traumatic Brain Injury in Underserved Populations Research Program for their feedback on this scoping review.

## Author Contributions

**Conceptualization:** Samira Omar, Vincy Chan.

**Formal analysis:** Samira Omar, Maria Jennifer Estrella, Vincy Chan.

**Funding acquisition:** Angela Colantonio, Vincy Chan.

**Methodology:** Samira Omar, Maria Jennifer Estrella, Jessica Babineau, Vincy Chan.

**Project administration:** Maria Jennifer Estrella, Vincy Chan.

**Resources:** Angela Colantonio.

**Supervision:** Samira Omar, Maria Jennifer Estrella, Angela Colantonio, Vincy Chan.

**Visualization:** Samira Omar, Maria Jennifer Estrella, Muzna Ahmad, Vincy Chan.

**Writing – original draft:** Samira Omar, Maria Jennifer Estrella, Vincy Chan.

**Writing – review & editing:** Samira Omar, Maria Jennifer Estrella, Muzna Ahmad, Angela Colantonio, Jessica Babineau, Vincy Chan.

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
