## [Decision Letter · Decision Letter 0]

24 May 2024

PONE-D-24-01772Rehabilitation and Violence-related Traumatic Brain Injury: A scoping reviewPLOS ONE

Dear Dr. Chan,

Thank you for submitting your manuscript to PLOS ONE. After careful consideration, we feel that it has merit but does not fully meet PLOS ONE’s publication criteria as it currently stands. Therefore, we invite you to submit a revised version of the manuscript that addresses the points raised during the review process.

We look forward to receiving your revised manuscript.

Kind regards,

Andreas K Demetriades, MBBChir, MPhil, FRCSEd, FEBNS.

Academic Editor

PLOS ONE

Journal Requirements:

Reviewers' comments:

Reviewer's Responses to Questions

**Comments to the Author**

1. Is the manuscript technically sound, and do the data support the conclusions?

Reviewer #1: Yes

2. Has the statistical analysis been performed appropriately and rigorously? 

Reviewer #1: N/A

3. Have the authors made all data underlying the findings in their manuscript fully available?

Reviewer #1: Yes

4. Is the manuscript presented in an intelligible fashion and written in standard English?

Reviewer #1: Yes

5. Review Comments to the Author

Reviewer #1: Dear Author

This study, a scoping review, identifies the current state of rehabilitation for violence-related traumatic brain injury.

Appropriate rehabilitation programs for survivors of violence-related TBI are critical and it requires a different perspective than the other causes of brain trauma in advancing rehabilitation.

There are only a limited number of research reports on TBI rehabilitation from SDoH perspective, so it is very significant to sort out what is clear or unclear at this point.

The Methods is adequately described and well considered. The terms are clearly defined and search procedures are appropriate. But I am concerned about the study

design of the incruded articles showed in table2. Considering the evidence level, you should mention this point. Especially, it contains 5 RCT. Isn’t it worth reviewing?

Therefore, I think the manuscript should be revised, let me know your thoughts.

6. PLOS authors have the option to publish the peer review history of their article (what does this mean?). If published, this will include your full peer review and any attached files.

Reviewer #1: No

---

## [Author Response · Author response to Decision Letter 0]

27 Aug 2024

Responses to Reviewer Comment: PLOS ONE

Dear Editors of PLOS ONE,

We wish to thank you and the reviewer for their interest in our manuscript entitled Rehabilitation and Violence-related Traumatic Brain Injury: A scoping review. We greatly appreciate the interest and time of the reviewers who provided thoughtful suggestions to strengthen this manuscript. Below please find the verbatim comments from reviewers followed by my point-by-point response for each comment. Revisions are incorporated in the main document using tracked changes.

Reviewer 1 Comment: Is the manuscript technically sound, and do the data support the conclusions? The manuscript must describe a technically sound piece of scientific research with data that supports the conclusions. Experiments must have been conducted rigorously, with appropriate controls, replication, and sample sizes. The conclusions must be drawn appropriately based on the data presented.

Reviewer #1 Response: Yes

Reviewer 1 Comment: Has the statistical analysis been performed appropriately and rigorously?

Reviewer #1 Response: N/A

Reviewer 1 Comment: Have the authors made all data underlying the findings in their manuscript fully available? The PLOS Data policy requires authors to make all data underlying the findings described in their manuscript fully available without restriction, with rare exception (please refer to the Data Availability Statement in the manuscript PDF file). The data should be provided as part of the manuscript or its supporting information, or deposited to a public repository. For example, in addition to summary statistics, the data points behind means, medians and variance measures should be available. If there are restrictions on publicly sharing data—e.g. participant privacy or use of data from a third party—those must be specified.

Reviewer #1 Response: Yes

Reviewer 1 Comment: Is the manuscript presented in an intelligible fashion and written in standard English? PLOS ONE does not copyedit accepted manuscripts, so the language in submitted articles must be clear, correct, and unambiguous. Any typographical or grammatical errors should be corrected at revision, so please note any specific errors here.

Reviewer #1 Response: Yes

Reviewer 1 Comment: Dear Author, This study, a scoping review, identifies the current state of rehabilitation for violence-related traumatic brain injury. Appropriate rehabilitation programs for survivors of violence-related TBI are critical and it requires a different perspective than the other causes of brain trauma in advancing rehabilitation. There are only a limited number of research reports on TBI rehabilitation from SDoH perspective, so it is very significant to sort out what is clear or unclear at this point.

Response: Thank you so much for this important comment. To address the objectives of the review, we paid closed attention to intersecting SDoH (e.g., sex, gender, race, ethnicity, age, and other sociodemographic characteristics) in the analysis of the articles and writing of the manuscript. However, please note that information on intersecting SDoH was reported and integrated into the analysis, as it was available in the articles reviewed. In light of your thoughtful comment, we added the following sentence to the results on lines 221-223, to account for the fact that only a small number of the articles actually considered race, sex, and/or gender in the analysis and reporting of their data: “Only four studies provided an intersectional analysis that was stratified by race and gender and accounted for racial disparities in rehabilitation outcomes for persons with violence-related TBI [39-41, 44].”

We also added the following statements (lines 495-497) to acknowledge the need for a trauma-informed care to rehabilitation for this population given intersecting social determinants of health: “Important to note is the lack of consideration for a trauma-informed approach to rehabilitation given intersecting SDoH and the need for trauma-informed rehabilitation care, particularly for this clinical population of individuals who sustained a violence-related TBI.” 

Reviewer 1 Comment: The Methods is adequately described and well considered. The terms are clearly defined and search procedures are appropriate. But I am concerned about the study design of the included articles showed in table2. Considering the evidence level, you should mention this point. Especially, it contains 5 RCT. Isn’t it worth reviewing? Therefore, I think the manuscript should be revised, let me know your thoughts.

Response: Thank you so much for this comment. This scoping review aimed at synthesizing the existing literature on rehabilitation for violence-related TBI drawing from peer-reviewed literature as well as grey literature reports. All studies included in this review including the 5 RCTs were analyzed and integrated in the results section. We added a clarifying statement on lines 205-206 of the results to clarify that these studies were reviewed and analyzed:“Five (8.8%) of these studies were RCT’s and three were specifically qualitative (5.3%) by design.”

Reviewer 1 Comments: PLOS authors have the option to publish the peer review history of their article (what does this mean?). If published, this will include your full peer review and any attached files.

---

## [Decision Letter · Decision Letter 1]

6 Sep 2024

Rehabilitation and Violence-related Traumatic Brain Injury: A scoping review

PONE-D-24-01772R1

Dear authors

We’re pleased to inform you that your manuscript has been judged scientifically suitable for publication and will be formally accepted for publication once it meets all outstanding technical requirements.

Kind regards,

Andreas K Demetriades, MBBChir, MPhil, FRCSEd, FEBNS.

Academic Editor

PLOS ONE

Additional Editor Comments (optional):

Reviewers' comments:

Reviewer's Responses to Questions

**Comments to the Author**

1. If the authors have adequately addressed your comments raised in a previous round of review and you feel that this manuscript is now acceptable for publication, you may indicate that here to bypass the “Comments to the Author” section, enter your conflict of interest statement in the “Confidential to Editor” section, and submit your "Accept" recommendation.

Reviewer #1: All comments have been addressed

2. Is the manuscript technically sound, and do the data support the conclusions?

Reviewer #1: Yes

3. Has the statistical analysis been performed appropriately and rigorously? 

Reviewer #1: N/A

4. Have the authors made all data underlying the findings in their manuscript fully available?

Reviewer #1: Yes

5. Is the manuscript presented in an intelligible fashion and written in standard English?

Reviewer #1: Yes

6. Review Comments to the Author

Reviewer #1: (No Response)

7. PLOS authors have the option to publish the peer review history of their article (what does this mean?). If published, this will include your full peer review and any attached files.

Reviewer #1: No

---

## [Editor Report · Acceptance letter]

5 Nov 2024

PONE-D-24-01772R1 

PLOS ONE

Dear Dr. Chan, 

I'm pleased to inform you that your manuscript has been deemed suitable for publication in PLOS ONE. Congratulations! Your manuscript is now being handed over to our production team.

Kind regards, 

on behalf of

Dr. Andreas K Demetriades 

Academic Editor

PLOS ONE